# Heavy Metals and Microbes Accumulation in Soil and Food Crops Irrigated with Wastewater and the Potential Human Health Risk: A Metadata Analysis

Yahia A. Othman [1,*], Amani Al-Assaf [2], Maher J. Tadros [3] and Abeer Albalawneh [4]

1   Department of Horticulture and Crop Science, The University of Jordan, Amman 11942, Jordan
2   Department of Agricultural Economics and Agribusiness, The University of Jordan, Amman 11942, Jordan; amani.alassaf@ju.edu.jo
3   Department of Natural Resources and Environment, Jordan University of Science and Technology, Irbid 22110, Jordan; mtadros@just.edu.jo
4   Environment and Climate Change Directorate, National Agricultural Research Center, Baqa, Amman 19381, Jordan; aberfer@yahoo.com
*   Correspondence: ya.othman@ju.edu.jo; Tel.: +962-779695781

**Abstract:** Wastewater is actively used for irrigation of vegetable and forage crops in arid lands due to water scarcity and cost advantages. The objective of this review was to assess the effect of wastewater (mixture sources) reuse in irrigation on soil, crop (vegetable and forage crops), animal products, and human health. The metadata analysis of 95 studies revealed that the mean of toxic heavy metals including nickel (Ni), chromium (Cr), cadmium (Cd), lead (Pb), and zinc (Zn) in untreated wastewater were higher than the world standard limits in wastewater-irrigated regions. Although heavy metals in treated wastewater were within the standard limits in those areas, the concentration of those toxic elements (Pb, Cd, Ni, Cr, and As) exceeded the allowable limits in both soil and vegetables' edible parts. In fact, the concentration of heavy metals in vegetables' edible parts increased by 3–9 fold when compared with those irrigated with fresh water. *Escherichia coli* in wastewater-irrigated soil was about $2 \times 10^6$ (CFU $g^{-1}$) and about 15 (CFU $g^{-1}$) in vegetables' edible parts (leaf, bulb, tuber and fruit) while the mean total coliforms was about $1.4 \times 10^6$ and 55 (CFU $g^{-1}$) in soil and vegetables' edible parts, respectively. For human health risk assessment, the estimated daily intake (EDI) and human health risk index (HRI) ranged from 0.01 to 8 (EDI and HRI > 1.0 associated with adverse health effects). Although the mean of EDI for heavy metals from wastewater-irrigated vegetables were less than 1, the HRI for Cd and Pb were above the limits for safe consumption. Overall, heavy metal levels in wastewater that used for irrigation of agricultural crops could be within the recommended levels by the world standards, but the long-term use of this reused water will contaminate soil and crops with several toxic heavy metals leading to potential carcinogenic risks to humans. Therefore, rigorous and frequent testing (wastewater, soil, and plant) is required in cultivated farms to prevent the translocation of heavy metals in the food chain.

**Keywords:** estimated daily intake; health assessment; forage crops; vegetable crops; toxic metals; *Escherichia coli*

## 1. Introduction

Water availability is a critical limiting factor for the sustainable development of arid lands which covers about 26% of the Earth's lands and accommodate 25% of the global population [1]. These arid regions are sensitive to global warming consequences, such as high temperature and frequent droughts. Drought is a key limiting factor for sustainable crop production in the arid and semi-arid lands [2]. In addition, rapid population growth of these regions, poor management practices as well as political turmoil have increased the pressure on the limited water resources [1,2]. For example, the civil war in Syria (2011–now), and the associated wave of refugees entering Jordanian lands (1.3 million,

10% of Jordanian population), have stressed the Jordanian economy as well as their water resources [2]. Therefore, governments in arid regions and the other countries with limited water resources have adopted several water management policies to sustain their water including rainwater harvesting and the reuse of wastewater in irrigation of agricultural crops [3,4].

Wastewater reuse in irrigation of agricultural crops is a common practice in several parts of the world, especially arid and semi-arid regions [5]. Wastewater is mainly used in irrigation, as it is used to irrigate 20 million ha in about fifty countries [6]. Interestingly, the total irrigated area prepared for direct use of raw or treated wastewater is estimated to be 8.4 million hectares in 42 countries [7]. The use of wastewater in irrigation is a common practice in developing countries (e.g., Jordan) as an adaptation strategy to sustain water resources [3,8]. Most water conservation approaches are expensive, ineffective, and may require expertise for identifying water challenges, especially in developing countries. Farmers prefer the reuse of wastewater for irrigating their crops due to (1) low wastewater price, (2) lower use of fertilizers (compared to fresh water) due to wastewater properties (22% Nitrogen, 14% phosphorus) [9,10], (3) expanding cultivated areas for crop [10,11], and (4) for being a reliable source of renewable energy. The reuse of wastewater in irrigation reduces the cost of crop production and helps in liberating the capital resources for more investments in agricultural projects, besides the contribution in the municipality costs of searching alternative water sources through adopting more developed and expensive means [12,13]. In addition, recycling wastewater for reuse in irrigation contributes for designing attractive investment policies and accessible financing mechanisms that aim to provide capital for preventing pollution and other health hazards associated with wastewater used in irrigation [13,14]. Considering the limited water resources due to changing climate and the exponential increase in population growth, the adoption of wastewater for irrigation of food crops and the sustainable conservation of water in arid land regions is inevitable [15].

Although wastewater is an essential source for irrigation of agricultural crops, potential environmental and human health risk issues are linked to the reuse of this type of water due to the accumulation of heavy metals (lead (Pb), cadmium (Cd), nickel (Ni), chromium (Cr) and zinc (Zn)) and microbes (*Escherichia coli*) [6]. In fact, accumulation of wastewater contaminants, including heavy metals in the environment is inevitable. The contamination of toxic metals negatively impacts all environmental constituents, specifically the aquatic ecosystems [16]. In Pakistan, industrial wastewater analyses reveled that total suspended solids (190%), Ni (16%), Cd (80%), and Pb (106%) were higher than the maximum permissible limits [17]. In addition, the hazard index and the carcinogenic and non-carcinogenic dermal health risks of irrigation with industrial wastewater was substantially higher than fresh water. Rivers, dams, and groundwater are the main sources of drinking water for developing countries. However, unrestricted reuse and disposal of domestic and industrial wastes allows those freshwater reservoirs to serve as the best sinks for the discharge of wastewater [18,19]. For example, In Iran, the low quality supply Sabalan dam increased heavy metals (As) accumulation in the reservoir and consequently increased the carcinogenic human health risk of using this contaminated water to $1.69 \times 10^{-4}$ [20]. Elements including essential minerals can cause toxic effects to plants, animals, and humans if present at excess concentrations [21]. Here, we analyzed the results of more than 90 studies that used wastewater for irrigation of vegetable and forage crops. We assessed the impact of wastewater on soil, animal health and products, especially dairy cows as well as human health. The metadata analyses included the levels of heavy metals (Pb, Ni, Cd, Cr, and Zn) and microbes (total coliforms, fecal coliforms, and *Escherichia coli*) in wastewater, irrigated-soil, and crops as well as human health risk.

## 2. Wastewater Quality for Reuse in Irrigation of Agricultural Crops

High demands for water in dry lands plus frequent drought encourage policymakers and governments to assess alternative management practices to sustain water resources and

achieve greater production [22]. Water scarcity and unreliable rainfall (Table 1) make water management practices, such as rainfall water harvesting and the use of non-conventional water of lower quality (saline and wastewater), viable investment for water supply and food production in drylands [3,4,23]. Sustainable land management requires comprehensive attention and a long-term vision to both bio-physical and socio-economic aspects [24]. The upgrade of water supply systems coupled with public awareness to reduce residential water use and saving as well as the reuse of domestic waste and greywater have been increased recently [3,25,26]. In fact, the direct use of treated and untreated municipal wastewater for irrigation purposes has been increased recently worldwide to compensate the water shortage crises. In 2018, the total annual wastewater ($10^9$ m$^3$/year) in Mexico was 5.71, India 1.23, China 1.26, and Jordan 0.1 (Table 1). Interestingly, several countries have developed wastewater irrigation systems for direct reuse. Worldwide, the total irrigated area equipped for direct use of wastewater is about 8.42 million ha; of this amount, untreated wastewater is 4.14 million ha and treated wastewater is 4.28 million ha [7].

**Table 1.** Total cultivated area, annual precipitation, total volume of wastewater used for irrigation and total area equipped for wastewater irrigation for some countries that use wastewater for irrigation [7].

| Country | Rain-Fed Cultivated Area (1000 ha) | Irrigated Cultivated Area (1000 ha) | Annual Precipitation in Volume ($10^9$ m$^3$/Year) | Annual Precipitation in Depth (mm/Year) | Wastewater (Treated and Non-Treated) for Irrigation ($10^9$ m$^3$/Year) | Total Irrigated Area Equipped for Direct Use of Treated Wastewater (1000 ha) |
|---|---|---|---|---|---|---|
| India | 76,742 | 92,575 | 3560 | 1083 | 1.23 | 1.32 |
| Pakistan | 12,710 | 18,590 | 393 | 494 | 1.02 | 32.5 |
| Iran | 8544 | 8893 | 86 | 228 | 0.33 | 240 |
| China | 40,190 | 95,486 | 6192 | 645 | 1.26 | 3618 |
| Australia | 29,008 | 2298 | 4133 | 534 | 0.14 | - |
| Japan | 1463 | 2957 | 630 | 1668 | 0.11 | - |
| Jordan | 197 | 83 | 9.9 | 111 | 0.103 | 3.7 |
| Palestine | 62 | 124 | 2.4 | 402 | 0.013 | - |
| Iraq | 3107 | 2143 | 93.7 | 216 | 1.08 | - |
| Saudi Arabia | 2641 | 954 | 127 | 59 | 0.53 | 51.92 |
| Turkey | 18,974 | 4206 | 466 | 593 | 0.05 | 9.16 |
| Bahrain | 4.0 | 0.6 | 0.06 | 83 | 0.009 | 1.25 |
| Algeria | 7658 | 858 | 212 | 212 | 0.01 | 1.2 |
| Egypt | 1339 | 2497 | 18.1 | 75 | 0.29 | 35.5 |
| Morocco | 7815 | 1711 | 155 | 346 | 0.01 | - |
| Argentina | 31,400 | 2301 | 1643 | 591 | 0.091 | 20 |
| Mexico | 16,276 | 6331 | 1489 | 758 | 5.71 | 70 |
| Bolivia | 4449 | 278 | 1259 | 1146 | 0.016 | 1.56 |
| Brazil | 55,107 | 8411 | 14,995 | 1761 | 0.008 | - |

Wastewater and greywater reuse in irrigation are essential sources for sustainable water management in arid lands, promoting the preservation of the limited freshwater resources [5,27]. Municipal wastewater is defined as water (99.9%) and suspended and dissolved organic (e.g., lignin, fats, soaps, synthetic detergents, proteins) and inorganic solids (heavy metals) collected from homes and industries (Figure 1). Greywater (50–80% of residential wastewater) is mainly comprised of water collected separately from sewage flow that originates from clothes washers, bathtubs, showers, and sinks, without wastewater from toilets [15]. The use of greywater for irrigation would reduce the demand on water resources and alleviate the pressure on wastewater treatment plants [15].

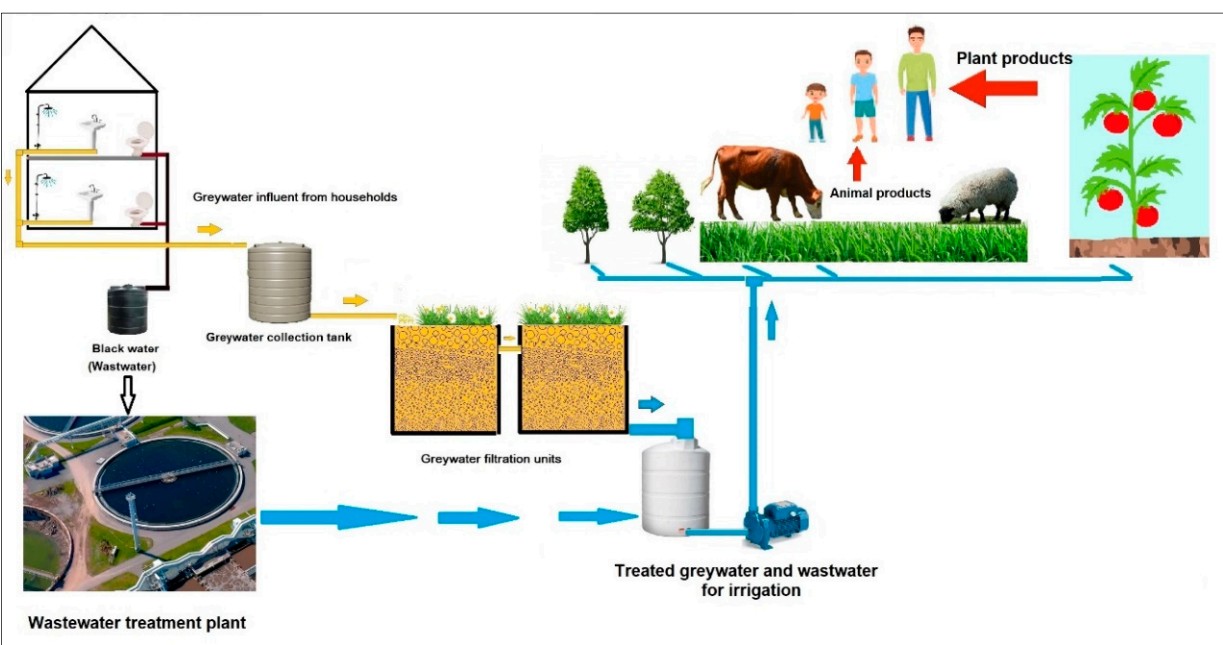

**Figure 1.** Schematic diagram for wastewater and greywater reuse in agriculture.

Numerous wastewater treatment and quality assessment methods have been used worldwide [16]. Microbial contaminants, such as total and fecal coliforms, normally exceed the recommended levels due to unrestricted entry of untreated wastewater into the environment; therefore, the use of suitable purification treatments with high removal efficiency for microbial agents are critical [28]. In addition, wastewater treatment plants might emit several microbiological contaminants in the air, including mesophilic, psychrophilic and coliform bacteria [29]. However, the abundance of these microbes in the air and the purified wastewater depends on the treatment plant capacity and purification system [29]. In Iran, the assessment of wastewater treatment systems (activated sludge, stabilization ponds, wetlands, and low and medium pressure UV disinfection systems) for microbial removal revealed that the active sludge systems was not efficient in reducing coliforms (total and fecal) compared to stabilization pond systems [30]. Udayanga et al. [31] found that thermal processing of sewage sludge, especially pyrolysis, valorized the carbon rich organic fraction of the sludge, while successfully reducing its volume. Ozonation-based disinfection methods can effectively remove antibiotic resistant bacteria from aqueous solutions [32]. However, the process efficiency was affected by the ozone dose as well as the wastewater solids and pH [32]. Biological treatments of wastewater normally remove microbial pollutants but these methods fail to eliminate numerous chemical compounds, such as pharmaceuticals [29]. Therefore, additional treatment process is necessary to remove these pollutants, including membrane filtration, adsorption, coagulation, electrochemical treatment, or advanced oxidation [29]. For example, adsorption onto activated carbons has been selected as the procedure to remove different chemical contaminant at the industrial scale [28]. Overall, wastewater treatment systems involve a combination of physical, biological, and chemical processes to purify the effluent efficiently [28].

The use of wastewater for irrigation of food crops is controversial. The economic and energetic assessment of wastewater reuse as viable complementary sources of water represent a possible opportunity [25]. In the arid regions, research studies have recommended the reuse (with precautions) of wastewater for irrigation in urban landscape and forage crop production [3,27,33]. However, analysis of this type of water depends on its source. The use of wastewater for irrigation might generate substantial environmental contamination and toxicology problem especially when farmers use untreated wastewater [6,34]. Wastewater contains toxic microorganisms and heavy metals such as Ni, Cd, Cr, and Pb that can induce severe risks to humans and the environment [6].

The chemical composition of tertiary treated wastewater are often within the World Health Organization (WHO) allowable limits, but heavy metal concentrations (i.e., Ni, Cd, Pb, As, and Cr) has exceeded the maximum limits in several regions of the world, especially in regions that use untreated wastewater for irrigation (Table 2). Microbial levels depend on wastewater sources and might contain pathogens, such as Pseudomonas, Salmonella, Aeromonas, and Staphylococcus; therefore, should not be reused for irrigation without treatment [35]. Table 2 shows that levels of total coliforms, fecal coliforms, and *Escherichia coli* have also exceeded the maximum world limits in untreated wastewater. In addition, treated wastewater microbial levels (total coliforms, fecal coliforms, and *Escherichia coli*) and some heavy metals (Cu, Cd, Cr, and Ni) were higher than the standard limits. Akoto et al. [34] found that irrigation of farmlands with untreated wastewater contaminated the soil by Ni, Pb, Cr, and Cd and transferred these contaminants to the farm products (lettuce). A study conducted by Qishlaqi et al. [5] showed that excessive accumulation of Ni and Pb was found in wheat tissues irrigated with untreated wastewater. They concluded that strict protection measures and rigorous integrated systems are essential to alleviate the negative effect of wastewater reuse in agriculture, especially the regions irrigated with untreated wastewater. The global assessment of irrigated croplands affected by urban wastewater (treated and untreated) revealed that about 36 million ha of irrigated croplands were located in wastewater dependent catchments [36]. In addition, about 82% of these affected croplands (29.3 million ha) were located in countries (China, India, Pakistan, Mexico, and Iran) in which less than 75% of wastewater is normally treated [36]. Considering the microbial and chemical analysis of untreated and treated wastewater exceeding the world standard limits in some regions of the world as well as the recommendations of previous studies, untreated wastewater should be restricted (wastewater treatment is essential) with a periodic water analysis to reduce the transfer of heavy metals to crops, animals, and humans.

Several studies have suggested greywater reuse as an alternative to wastewater [25,26,35]. However, heavy metal levels of greywater and wastewater could be similar [3,37]. In addition, the collection of used water from kitchen and laundry only required extra cost and could be inapplicable in some residential regions. Therefore, the greywater reuse in irrigation is limited worldwide when compared to wastewater. When wastewater is considered for reuse, it is normally tested according to specific quality standards to avoid threatening the environment and humans [38,39]. These water quality standards consist of chemical and microbiological components. However, only specific variables are measured and the probability of having non-measured toxic substances is high [38,40]. In Australia, 22 organic micro-pollutants including triclosan, caffeine, paracetamol, acesulfame, and salicylic acid were found in greywater and thus the reuse of this recycled-irrigation water can act as a source of microbial pollutant to soil, plants, and groundwater [41]. Similarly, in Palestine, the antibiotic and herbicide analysis of residential greywater used for irrigation of food crop production showed that antibiotics and herbicides were presented in the reused greywater. Those toxic antibiotics (1.3 to 1592.9 ng $L^{-1}$) and herbicide (3.1–22.4 ng $L^{-1}$) materials included tetracycline, ciprofloxacin, atrazine, erythromycin, oxolinic acid, and trifluralin [40].

**Table 2.** Quality of un-treated and treated wastewater compared with fresh and allowable word standard limit for irrigation. World standards represent the range limits for the WHO and the following countries: Canada, India, Jordan, Italy, Australia, Japan, China, Slovenia, Germany, and Great Britain. Bold values indicate where wastewater variable exceeded the standards limits.

| Parameter | Symbol and Unit | Untreated Wastewater | Treated Wastewater | Fresh Water | Word Standards | Reference |
|---|---|---|---|---|---|---|
| Potential of Hydrogen (H$^+$) | pH | 5–10 | 6–8 | 7.1–7.6 | 5.5–9.5 | [3,5,15,25,42,43] |
| Electrical conductivity | EC$_w$ (dS m$^{-1}$) | 0.5–10 | 0.4–0.8 | 0.3–4.0 | 0.7–10 | [3,15,25,42,43] |
| Total Dissolved Solids | TDS (mg L$^{-1}$) | 279–**2444** | 100–429 | 80–154 | 450–2000 | [3,15,44] |
| Total Suspended Solids | TSS (mg L$^{-1}$) | 2–987 | 2–312 | 0–21 | - | [25,42,45] |
| Sodium Adsorption Ratio | SAR | 2–**90** | 1–**21.9** | 3.0–8.0 | 9–13 | [3,15,44] |

**Table 2.** *Cont.*

| Parameter | Symbol and Unit | Untreated Wastewater | Treated Wastewater | Fresh Water | Word Standards | Reference |
|---|---|---|---|---|---|---|
| Turbidity | T (NTU) | 3–**444** | - | Not detected | 1–10 | [15,25,42,45] |
| Biological Oxygen Demand (5 days) | $BOD_5$ (mg $L^{-1}$) | **135–4450** | 10–**942** | 0.0–225 | 60–300 | [3,5,15,25,42,45,46] |
| Chemical Oxygen Demand | COD (mg $L^{-1}$) | 15–**4155** | 5–**1700** | Not detected | 120–500 | [3,15,25,42,45,46] |
| Fat, Oil and Grease | FOG (mg $L^{-1}$) | 8–**232** | - | Not detected | 8 | [15,45] |
| Anionic surfactants | ASR (mg $L^{-1}$) | 1–80.5 | 0.3–1.0 | Not detected | 0.1–100 | [15,25,45] |
| Methylene Blue Active Substances | MBAS (mg $L^{-1}$) | 1.6–**118** | 0.3–**39** | Not detected | 25 | [15] |
| Total Coliforms | TC (CFU 100 mL$^{-1}$) | 1000–**1.9 × 10$^8$** | 200–**2 × 10$^7$** | 0.0–2.0 | 10–1000 | [15,25,44,45] |
| Fecal Coliforms | FC (CFU 100 mL$^{-1}$) | 200–**2 × 10$^7$** | 10–**4 × 10$^6$** | 0.0–2.0 | 2–1000 | [15,25,44,45] |
| *Escherichia coli* | *E. coli* (CFU 100 mL$^{-1}$) | 1000–**8 × 10$^8$** | 408–**4 × 10$^5$** | 0.0–1.0 | 1000–10$^5$ | [3,15,25,44,45] |
| Bicarbonate | HCO$_3$ (mg $L^{-1}$) | 2–223 | - | - | 520 | [15,45] |
| Orthophosphate | PO$_4$ (mg $L^{-1}$) | 0–**52** | 0.2–3.2 | 0.03–0.8 | 30 | [43,47] |
| Nitrate | NO$_3$ (mg $L^{-1}$) | 10–**52** | 0.1–7.0 | 0.0–0.004 | 45–50 | [15,25,42,43,45] |
| Sulfate | SO$_4$ (mg $L^{-1}$) | 1–22 | 0.5–28 | 0.0–0.1 | 500 | [15,41,43–45] |
| Total nitrogen | TN (mg $L^{-1}$) | 1–**61** | 0.5–17.7 | 2.0–10 | 5–50 | [3,15,25,33,43,45,46] |
| Potassium | K$^+$ (mg $L^{-1}$) | 20–39 | 1–10 | 0.0–12 | 80 | [3,5,15,33,43,44] |
| Phosphorus | P (mg $L^{-1}$) | 26–38 | 0.05–1.2 | 0.01–1.0 | - | [3,5,33,44–46] |
| Calcium | Ca$^{+2}$ (mg $L^{-1}$) | 1–100 | 0.1–72 | 1.5–15 | 230–400 | [3,5,15,33,44,45] |
| Magnesium | Mg$^{+2}$ (mg $L^{-1}$) | 1–60 | 0.1–23 | 0.0–10 | 60 | [3,5,15,33,44,45] |
| Manganese | Mn (mg $L^{-1}$) | 0.02–0.16 | 0.0002–0.06 | 0.0–0.17 | 0.2 | [5,15,46,48–50] |
| Iron | Fe (mg $L^{-1}$) | 0.1–2.7 | 0.1–0.4 | 0.0–0.1 | 0.1–5 | [3,5,15,33,46,48,49] |
| Zinc | Zn (mg $L^{-1}$) | <0.002–**13.0** | 0.01–0.7 | 0.0–0.17 | 2.0 | [3,5,15,48,49] |
| chloride | Cl$^-$ (mg $L^{-1}$) | 9–**450** | 63–205 | 1.0–18 | 140–400 | [3,15,45,46] |
| Sodium | Na$^+$ (mg $L^{-1}$) | 2–**667** | 1.0–136 | 2.0–19 | 69–230 | [3,15,44] |
| Copper | Cu (mg $L^{-1}$) | 0.001–**91** | 0.001–**24** | 0.001–0.02 | 0.2 | [15,46,48,49] |
| Boron | B (mg $L^{-1}$) | 0.02–0.44 | 0.001–0.04 | 0.0–0.1 | 0.7–3.0 | [3,15,25,45,51] |
| Aluminum | Al (mg $L^{-1}$) | 0.0–**21** | 0.0–1.5 | 0.0–0.03 | 5.0 | [3,15,44,45,51] |
| Cadmium | Cd (mg $L^{-1}$) | <0.001–**4.0** | <0.002–**0.4** | 0.0–**0.03** | 0.01 | [3,5,15,48–50,52] |
| Lead | Pb (mg $L^{-1}$) | <0.003–**84** | <0.01–1.3 | 0.003–5.0 | 5.0 | [3,5,15,48–50,52] |
| Chromium | Cr (mg $L^{-1}$) | <0.004–**42** | 0.0–**4.0** | <0.008–0.8 | 0.1 | [5,15,45,48,50–52] |
| Arsenic | As (mg $L^{-1}$) | 0.0001–**6.0** | 0.001–0.002 | <0.0025 | 0.1 | [3,15,44,45,48,51] |
| Nickel | Ni (mg $L^{-1}$) | 0.04–**70** | 0.01–**9.0** | <0.001–**6.0** | 0.2 | [5,15,33,48–50] |
| Cobalt | Co (mg $L^{-1}$) | 0.0–0.01 | 0.0–<0.0001 | Not detected | 0.05 | [44–46,51] |
| Selenium | Se (mg $L^{-1}$) | <0.001 | <0.001 | Not detected | 0.02 | [45] |
| Vanadium | V (mg $L^{-1}$) | 0.0001–0.004 | - | Not detected | 0.1 | [51] |
| Mercury | Hg (mg $L^{-1}$) | 0.001 | - | Not detected | 0.0001–0.01 | [52] |

## 3. Effect of Wastewater Reuse on Soil and Plant

The use of wastewater in arid lands for irrigation of vegetable and forage crops is common due to the high salinity of overexploited aquifers, water scarcity, and cost considerations [34,53]. In many arid and semi-arid regions (e.g., Jordan, Iraq, Ghana Saudi Arabia, and India), irrigation is essential to overcome the prolonged drought periods during the summer. In salt-affected arid lands (due to overexploitation of aquifers), farmers are turning to wastewater as a source of low saline water [53]. The reuse of wastewater for irrigation may have several beneficial effects for plants because it increases the levels of some beneficial elements (N, P, K, Fe, Zn, Ca, and Mg) in the soil [53]. Table 2 shows that micronutrients concentration (N, P, K, and Ca) in untreated wastewater are much higher than fresh water. Higher nutrient and organic matter in the soil lead to a higher growth rates and production. The use of treated wastewater in irrigation effectively increased stem height and the dry matter of *Panicum maximum* compared to those irrigated with fresh water [53]. The higher performance of the wastewater treatment can be explained by its higher nutritive content, especially in N [53].

Heavy metals such as Cd, Cr, Pb, and Ni are metallic elements, have relatively higher weight than water, are extremely soluble in the aquatic environments, and consequently, they can be uptake easily by living organisms (plant, animal, human) [52]. Turner et al. [51] found that the use of greywater for irrigation gradually increased soil (B, Cr, As, and Cu) and groundwater metals (Al, As, Cr, Cu, Fe, Mn, Ni, and Zn) exceeded safe limits standards after four years.

Vegetables are an essential component of our daily diet. However, the ability of vegetable growers to provide the ever-growing population with the required amount is limited by the unpredicted rainfall pattern and unsuitable irrigation systems [34]. Therefore,

farmers tend to use wastewater, which is readily available alternative for irrigation in most dryland regions. However, wastewater contains a substantial amount of pollutants, such as heavy metals. Figure 2 shows the metadata analysis results of previous studies that assess the accumulation of heavy metals in wastewater irrigated vegetables; leafy-green (lettuce, spinach, parcel, mint, cabbage, pudina, and coriander), bulbs and tubers (onion, garlic, potato, radish, and carrot), and fruits (tomato, pepper, cauliflower, okra, and eggplant). Although heavy metals in wastewater were within the standard limits, the concentration of those toxic elements (Pb, Cd, Ni, Cr, As, and Zn) exceeded the allowable limits in both soil and vegetables edible parts (Figure 2). In fact, the concentration of heavy metals in vegetable edible parts increased by 3–9 fold compared to those irrigated with fresh water. For example, leafy-green from wastewater-irrigated fields increased Pb concentration by 5 fold, Cd by 7 fold, Ni by 8 fold, Cr, As, and Zn by about 6 fold compared to those irrigated with fresh water. Khan et al. [50] used wastewater for irrigation of vegetable crops (spinach, coriander, carrot, tomato, and cauliflower). They found that all tested leafy-green, root, and fruit vegetable samples were contaminated with high levels of Pb, Ni, and Cd; higher than WHO limits. Qureshi et al. [42] found that the concentration of Zn and Cr in leafy-green vegetables (lettuce and spinach) was higher than root and fruit (tomato, eggplant, radish, and carrots) vegetables. Therefore, selecting suitable crops can potentially reduce the health risk for humans. In this review, the metadata analysis showed that the concentrations of Pb and As were similar across vegetable types while leafy-green had higher Ni, Cr, and Zn than bulb, tubers, and fruit vegetables (Figure 2). Overall, although the concentration of heavy metals in wastewater used for irrigation were within the WHO limits, the long-term reuse of this recycled water led to excessive build-up of those toxic metals in the soil. Therefore, rigorous and continuous testing (wastewater, soil, plant) is required in cultivated farms to prevent the translocation of heavy metals in the food chain [54].

Soil is the key component for developing an integrated and sustainable wastewater management system. This is because the chemistry and physics of the soil can significantly affect the levels of toxic materials in the soil and consequently the quality of the crops. The reviews of previous studies, conducted on wastewater reuse in agriculture, revealed that most studies that positively recommended the reuse of wastewater were (1) short-term studies (less than 4 years) or/and (2) assuming that the analysis of wastewater only is sufficient for safe use in agriculture and thus could maintain the level of heavy metals in the crops within the recommended WHO limits [3,33,43,46]. However, long-term studies on wastewater reuse found that several heavy metals significantly increase across years leading to potential soil contamination. The long-term irrigation (~20 years) of wastewater in Shiraz, Iran increased organic matter of the soil by 20–30%, pH by 2–3 units, and heavy metals levels by more than 100%; exceeded the WHO limits [5]. Although the wastewater quality was acceptable in that study (Ni 0.19; Zn 0.06; Cd 0.004; Pb 0.33; Cr 0.1 mg kg$^{-1}$), the frequent irrigation led to accumulation of contaminated soil in the top 10 cm soil; Pb 441 mg kg$^{-1}$ (soil limits: 20–70 mg kg$^{-1}$), Cd 3.2 mg kg$^{-1}$ (limits: 0.5–1.0 mg kg$^{-1}$), Ni 297 mg kg$^{-1}$ (limits: 15–20 mg kg$^{-1}$), Cr 29 mg kg$^{-1}$ (limits: 30–50 mg kg$^{-1}$) and Zn 170 mg kg$^{-1}$ (limits: 60–100 mg kg$^{-1}$) [5]. In Nigeria, the vertical distribution analysis and modeling of heavy metals in vegetable farms irrigated with wastewater showed that these gardens will not be suitable for human consumption after 10–20 years if the heavy metal balances (input from wastewater and output metal taken out by plant biomass or leaching) remain unchanged [55].

Heavy metals accumulation potential is different between plant species. The transfer factor is normally used to estimate the translocation of those toxic metals from the soil to plant species [56]. The transfer factor is the ratio between the heavy metal concentrations in the edible part of vegetables (mg kg$^{-1}$) to the concentration of the metal in soil. Interestingly, Meng et al. [56] found that the transfer factor of heavy metals (especially, Cd and Pb) from soil to vegetables was extremely high. For example, the transfer factor of Cd for cabbage was 1.82 and for potato was 1.52. Similar results were found by Tiwari et al. [48] who found

that transfer of toxic metals (As, Cd, Cr, Pb, and Ni) from soil to edible parts of vegetables (pepper, cabbage, spinach, radish, and tomato) was high and unsafe due to possible transfer in the food chain leading to health hazards for humans. They suggested that only vegetable crops that restrict heavy metals in non-edible ports may be cultivated [48]. Overall, to guarantee food safety and the safe use of wastewater for irrigation, urgent attention is necessary to apply appropriate permanent monitoring and pollution control [56].

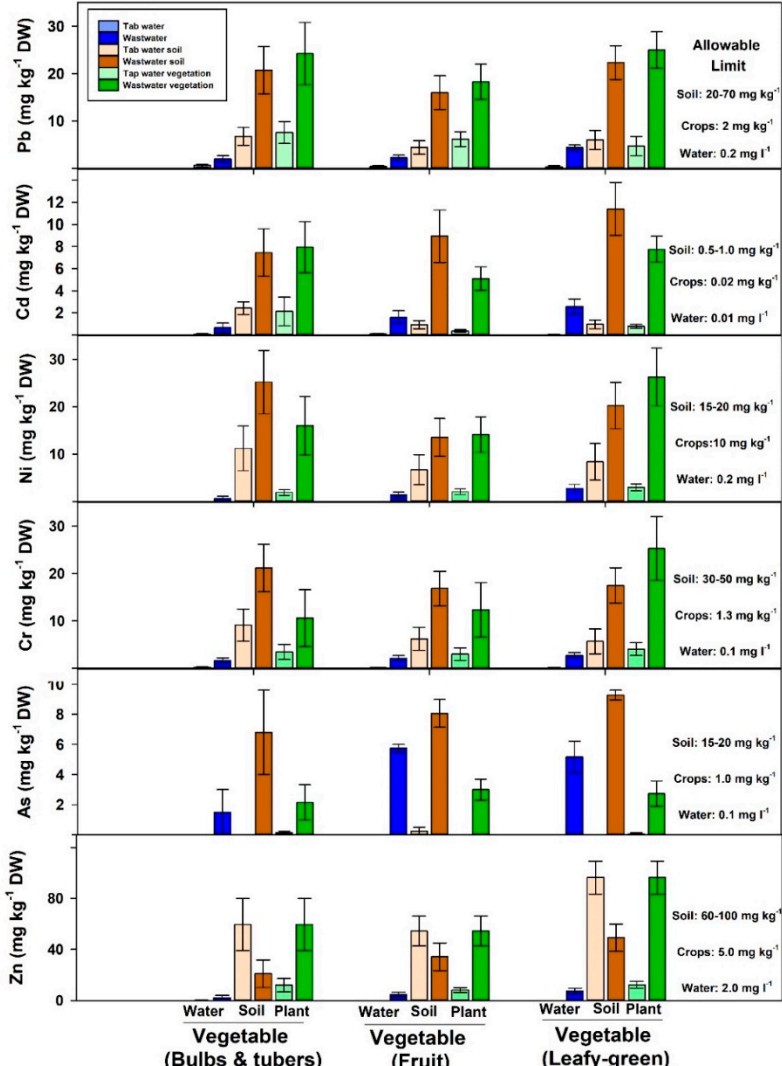

**Figure 2.** Heavy metals concentration in fresh and wastewater soil and vegetables [27,34,42,43, 46,48–50,53,57–66]. Leafy-greens represent lettuce, spinach, parcel, mint, cabbage, pudina, and coriander; bulbs and tubers represent onion, garlic, radish, potato and carrot; fruits vegetables represent tomato, pepper, cauliflower, okra, and eggplant. Bars represent mean ± SE.

Although treated wastewater is bacterially safe and has a positive impact on plant growth, microbiological contamination of vegetable crops irrigated with wastewater has been reported in many regions of the world [67,68] *Escherichia coli* is a coliform group of bacteria that is used to represent the bacterial pathogens in the reused wastewater and their behavior is expected to reflect enteric pathogens [15]. In this review, the mean total *Escherichia coli* in wastewater-irrigated soil was found to be about $2 \times 10^6$ (CFU $g^{-1}$) and about 15 (CFU $g^{-1}$) in vegetable edible parts (leaf, bulb, tuber, and fruit) (Figure 3). In addition, the mean total coliforms were about $1.4 \times 10^6$ (CFU $g^{-1}$) and about 55 (CFU $g^{-1}$) in vegetable edible parts (Figure 3). Qureshi et al. [42] found that all vegetables irrigated with wastewater had different levels of microbial loading in their edible part. The highest

level of contamination of total coliform were found in spinach, radish, and eggplant while the lowest concentrations were found in lettuce and tomatoes. In addition, the *Escherichia coli* counts on lettuce, tomatoes, eggplant, and carrot were higher than spinach. They concluded that the tertiary-level of wastewater treatment does not fully remove pathogenic bacteria (total coliform and *Escherichia coli*) from the reused wastewater nor the edible parts of the vegetables [42].

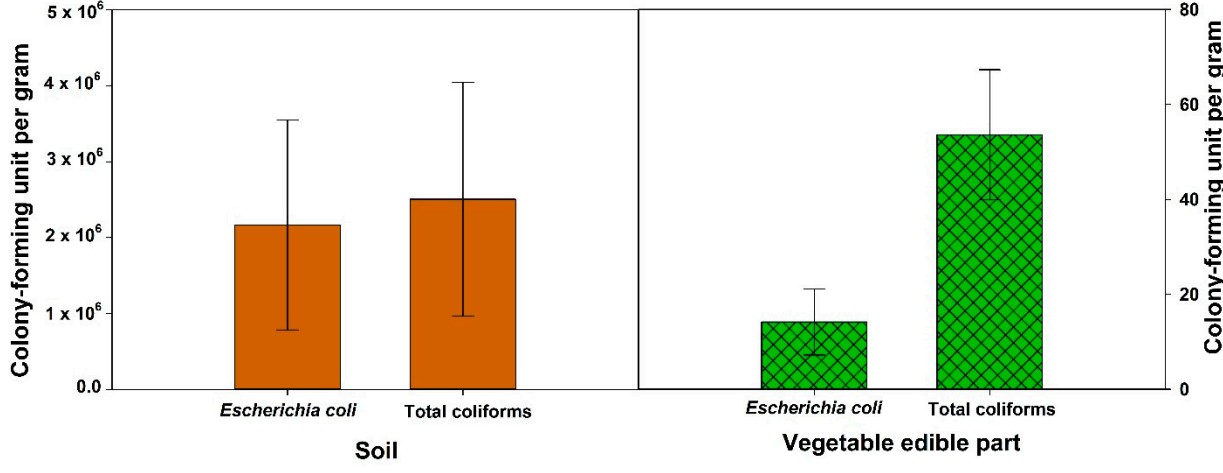

**Figure 3.** Microbial pathogen (*Escherichia coli* and total coliforms) levels in soil and vegetables irrigated with wastewater [42,44,67–71].

## 4. Wastewater Reuse for Animal Production

Forage production and feed quality are essential for providing healthy animal products for humans. Therefore, it is crucial to evaluate its safety in terms of impurities present in feeds [72]. Monitoring of toxic metals is one of the most important issues of sustaining feed quality [72]. Intensive industry and the use of advanced agricultural technology and low-quality water in forage production have emitted a wide variety of dangerous substances-pollutants to environment [73]. Consequently, several heavy metals have increased in animal feed [73]. Heavy metals, including Zn, Cu, Cr, As, Cd, and Pb are critical bioaccumulative toxins in the dairy production system [73]. When animals (e.g., dairy animals) consume feed contaminated with undesirable toxic elements, the contaminants may transfer to the animal source foods, such as liver, kidney, meat, and milk, and exert adverse effects on domestic animals [74–76]. In addition, toxic elements, such as Cd, cause several health problems to humans, including cancer, mutations, and fetal death [73]. Table 3 shows the concentration of heavy metals in dairy herd feed rations from more than 500 dairy farms in the USA, Italy, Netherlands, and Poland. In addition, heavy metals in animal body (blood, muscles tissues), and feces as well as in forage crops irrigated with wastewater are also presented (Table 3). The average content of heavy metals in dairy cattle diets were found to be within the maximum permitted levels, except for Zn. A high level of Zn in animal diets might be because it plays key roles in the activity of several enzymes in the animal body; therefore, Zn is included as a supplement for animals' diet, especially mineral mix [77]. Li et al. [74] found that the lowest concentration of toxic metals was found in plant-based forage (*Medicago sativa*) while the highest concentration was found in processed forage, specifically, those with mineral supplements [74]. Interestingly, field pea mixed with oat potentially decreased heavy metal content in green fodder [72]. Overall, the presence of toxic metal concentrations in the main feedstuff components is inevitable; however, management practices can reduce livestock metal exposures to these toxins and can limit metal transfer to human foodstuffs as much as possible [21].

**Table 3.** Concentration of heavy metals (mg kg$^{-1}$ dry weight) in dairy herd feed rations ($\pm$ standard deviation), animal body (blood, muscles tissues), and feces as well as in forage crops irrigated with wastewater [16,33,46,61,63,72–75,77–81]. Diary feed rations are from dairy farms in USA, Italy, Netherlands, and Poland. Bold values indicate where a trigger value was exceeded.

| Source | Zn | Ni | As | Pb | Cr | Cd |
|---|---|---|---|---|---|---|
| Animal feed irrigated with tap water | | | (mg kg$^{-1}$ dry weight) | | | |
| Alfalfa hay | 28.4 ± 9.5 | 2.20 ± 1.1 | - | 0.14 ± 0.06 | - | 0.075 ± 0.007 |
| Complementary mineral feed/Mineral mix | **3110** ± 2035 | - | 2.80 ± 1.5 | 2.03 ± 1.1 | - | 0.640 ± 0.09 |
| complete dairy rations | 80.7 ± 25 | - | 0.67 ± 0.4 | 0.50 ± 0.2 | 1.20 ± 0.7 | 0.078 ± 0.04 |
| Barley/corn/wheat grain or hay | 28.3 ± 11 | 3.65 ± 0.5 | 0.16 ± 0.09 | 0.53 ± 0.3 | | 0.08 ± 0.02 |
| Legume and grasses mixture | **224** ± 120 | 2.85 ± 0.9 | 0.63 ± 0.3 | 9.25 ± 3.7 | 3.05 ± 1.3 | 2.72 ± 1.1 |
| Soybean protein mix/soybean meal | 155 ± 51 | - | - | 0.87 ± 0.1 | - | 0.06 ± 1 |
| Concentration in animal faeces | **460** ± 340 | 37 ± 2.6 | 2.9 ± 0.4 | 20.7 ± 12.5 | 17.6 ± 10 | 2.4 ± 1.3 |
| Concentration in animal blood | 20 ± 1.8 | | 0.14 ± 0.1 | 0.63 ± 0.1 | | 0.0035 ± 0.0006 |
| Concentration in animal muscles tissues | 38.0 ± 2 | 0.01 ± 0.002 | - | 0.017 ± 0.01 | 0.015 ± 0.006 | 1.27 ± 0.25 |
| Forage crops irrigated with wastewater | 55.5 ± 35.4 | 3.1 ± 1.5 | 0.08 ± 0.04 | 18.8 ± 11.5 | 2.3 ± 1.5 | 0.33 ± 0.1 |
| Maximum acceptable concentration in animal feed | 120–180 | 10–50 | 2.0–10 | 10–30 | 100–1000 | 0.5–5.0 |

The soil-plant-animal relationships need to be considered for a better understanding of the metal impact on grazing animals (e.g., cows). The transfer of these toxic metals (e.g., Cd, Pb, Ni, and As) from soil to the animal's body occurs through plants that are the most important link in the food chain [72]. When plant roots accumulate in these toxic metals and deposit them in the vegetative parts, the contaminates become readily available for grazing animals [3]. High levels of heavy metals in forage plants destined for animal feed may lead to animal-derived products (e.g., milk and meat) being contaminated by the metals [72]. Previous work showed that heavy metals concentration in forage crops irrigated with wastewater were within the acceptable range for safe use in animal feed (Table 3). This is partially attributed to the lower irrigation frequency of forage crops compared to vegetables. In addition, field crops might have exclusion mechanisms that concentrate high amounts of heavy metals in the root system, which is not edible for grazing animals. However, a high amount of toxic metal in the soil will potentially increase the transfer of those elements to the shoots. Rusan et al. [81] studied the long-term (10 years) effect of wastewater irrigation on soil and forage (barley) quality parameters. Wastewater-irrigated barley had higher biomass and essential nutrients (Total-N, NO$_3$, P, and K) compared to the field irrigated with fresh water. However, Pb and Cd shoot concentrations were higher than non-treated plants. They concluded that proper management of wastewater irrigation, including periodical testing of soil and plant metals concentration, are required to ensure successful, safe, and sustainable long-term wastewater irrigation [81].

## 5. Human Health Risk Assessment of Wastewater-Irrigated Crops

The tendency of agricultural crops (vegetables, field crops, and forage) to accumulate heavy metals in their tissues is a public health concern because they are toxic to human health [82]. Heavy metals, including Cd, Pb, Hg, and As, are readily transferred through food chains and cause significant risk to animal and human health [50,82]. Several health organizations such as World Health Organization and International Agency for Cancer Research have declared As, Cd, Cr, Ni, and Pb to be human carcinogens [42,82]. Typically, the low doses of heavy metals found in vegetables has insignificant health risks for humans [42]. However, wastewater reuse in irrigation might lead to the accumulation of those toxic heavy metals in the soil, groundwater, and plants, thus causing potential health risks to consumers.

Human risk assessment in the food chain is critical in developing countries (e.g., India, Pakistan) where wastewater reuse in irrigation of agricultural crops is still unchecked [50]. Human health risk assessment is normally determined through calculating the health risk index (HRI) and daily intake of metal [82]. The estimated daily intake (EDI) of heavy metals is the ratio between metal concentrations in crops and the amount of consumption of the respective food crop, EDI = (concentrations of heavy metals in crops (mg kg$^{-1}$) × daily average intake of vegetables)/body weight. The average daily vegetable intake for adults is about 0.345 kg [82]. The HRI depends on daily intake of trace metals through oral

consumption of vegetables. Generally, the index value for EDI and HRI higher than 1.0 are associated with adverse health effects [42]. Higher values of metal pollution index and HRI indicates that heavy metal pollution in the site irrigated with wastewater presented a potential threat to human health [50]. The health risk assessment of heavy metals via crops intake from the wastewater irrigated sites of a dry tropical area of India showed that HRI was more than 1 for Cd and Pb in wheat, rice, eggplant, cauliflower, and cabbage [62]. In Pakistan, the EDI of heavy metals from vegetable crops (onion, garlic, tomato, and eggplant) irrigated with wastewater were nearly free of risks [83]. However, the other sources of toxic metal exposure, such as dermal contact and dust inhalation, have not yet been considered and might be critical [83].

The assessment of human health risk is a key procedure for hazardous substance management, planning remediation policies, and applying control measures [82,84]. Figure 4 shows EDI and HRI (adults) values of heavy metals (mg/kg/person/day) through consumption of vegetables irrigated with wastewater. This study shows that the EDI values for heavy metals from vegetable crops irrigated with wastewater were higher than those from clean-water-irrigated-vegetables. Although EDI of heavy metals from wastewater-irrigated-vegetables was less than one, the HRIs for Cd and Pb were above the limits for safe consumption (Figure 4).

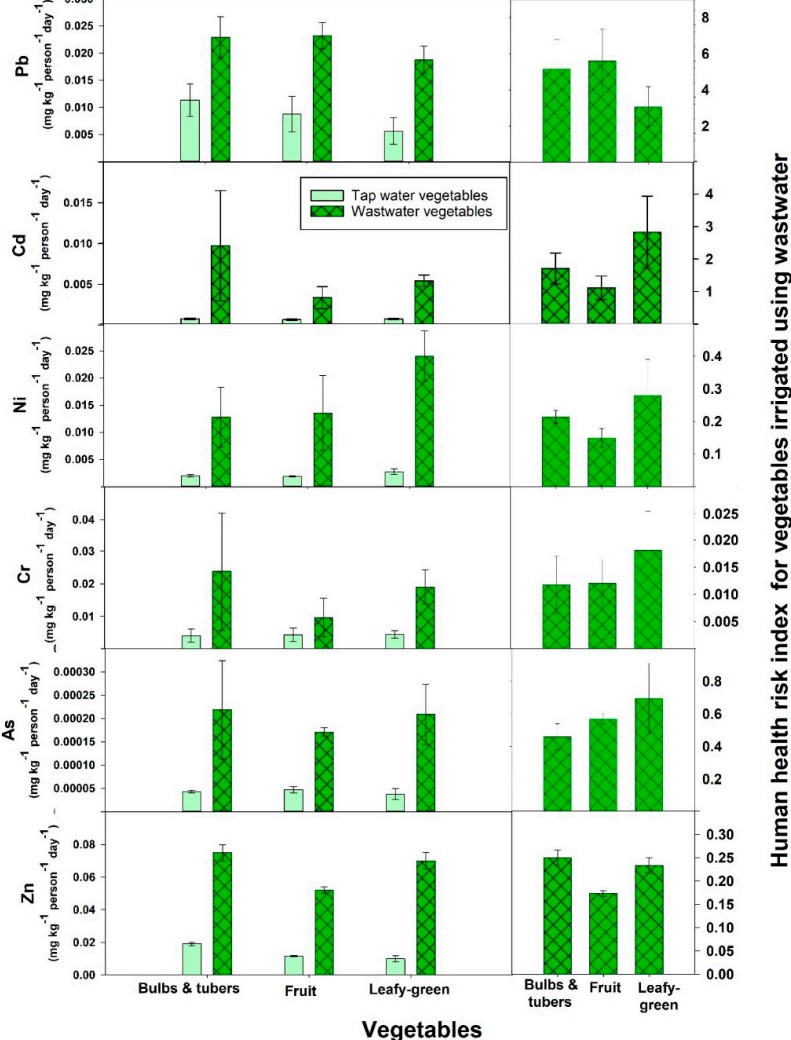

**Figure 4.** Estimated daily intake and human health risk index (adults) of heavy metals (mg/kg/person/day) through consumption of vegetables irrigated with wastewater [5,6,42,50,58,62,65,66,85–95]. A health risk index >1 for any metal in food crops means that the consumer population faces a health risk. Bars represent mean ± SE.

## 6. Conclusions

The use of untreated and treated wastewater for irrigation of agricultural crops (vegetables, field, and forage crops) is a common practice in arid and semi-arid regions. The total area equipped for irrigation by direct use of wastewater is about 8.42 million ha; untreated wastewater account for 49% at 4.14 million ha and treated wastewater accounts for 4.28 million ha. The metadata analysis showed that the concentration of heavy metals (Ni, Cd, Pb, Cr, and Zn) and microbial pathogens (total coliforms, fecal coliforms, *Escherichia coli*) in untreated wastewater were higher than the world standard limits. Although the heavy metals of treated wastewater were within the world limit for safe use in agriculture, the long-term reuse of treated wastewater leads to excessive build-up of those toxic metals in soil and crops. Wastewater quality standards for safe use in agriculture practices, such as those suggested by WHO, consist of chemical (heavy metals and nutrients) and micro-biological parameters (pathogens). However, wastewater sources might contain several non-measured toxic substances, which are not listed in the WHO standard tests, such as triclosan, paracetamol, acesulfame, and herbicides (e.g., atrazine, ciprofloxacin, and erythromycin). The metadata analysis showed that the concentration of heavy metals in vegetable's edible parts increased by 3–9 fold compared to those irrigated with fresh water. Consequently, the HRI for Cd and Pb in vegetable crops was above the limits for safe consumption. Given that the microbial and chemical analysis of untreated wastewater substantially exceeded the world standard limits for safe use, wastewater treatment should be a prerequisite for reuse in irrigation of agricultural crops. In addition, urgent attention is required to apply appropriate permanent monitoring and pollution control; a periodical water analysis should be carried out frequently to reduce the transfer of heavy metal to crops, animals, and human.

**Author Contributions:** Conceptualization, Y.A.O. and M.J.T.; formal analysis, A.A.-A., A.A. and Y.A.O.; writing—original draft preparation, Y.A.O., A.A., A.A.-A. and M.J.T.; writing—review and editing, Y.A.O. All authors have read and agreed to the published version of the manuscript.

**Funding:** This research received no external funding.

**Institutional Review Board Statement:** Not applicable.

**Informed Consent Statement:** Not applicable.

**Data Availability Statement:** Not applicable.

**Conflicts of Interest:** The authors declare no conflict of interest.

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
