# Peer review of "Heavy Metals and Microbes Accumulation in Soil and Food Crops Irrigated with Wastewater and the Potential Human Health Risk: A Metadata Analysis"

_water, doi:10.3390/w13233405_

Round 1

Reviewer 1 Report

Review report - water-1462237-peer-review-v1

In the manuscript, "Heavy metals and microbes accumulation in soil and food crops irrigated with wastewater and the potential human health risk: A metadata analysis", the authors conducted a brief but important review covering the effect of wastewater reuse in irrigation on soil, crop (vegetable and forage crops), animal products, and human health. Since wastewater has been widely used for irrigation of vegetables and crops in developing countries, a well-timed review could provide potential readers with the information about assessment and management of risks associated with wastewater irrigation. However, the current review lacks comprehensive, and only a limited number of research works have been cited. Meanwhile, it is also suggested that some contexts should be reorganized more logically. In addition, significant editing of the English is needed. If the revision is performed satisfactorily, the manuscript should be acceptable for publication.

Specific comments:

  1. Section 2 Wastewater quality for reuse in irrigation of agricultural crops: As wastewater treatment is essential, it is also necessary to review the wastewater treatment methods.
  2. Line 132-134: “The chemical composition (Ni, Cd, Pb, Fe, N, P, K) of tertiary treated wastewater are often within the World Health Organization (WHO) allowable limits, but heavy metal concentrations have exceeded the maximum limits in several regions of the world, especially, regions that use untreated wastewater for irrigation (Table 2).” Ni, Cd, Pb, Fe, N, P and K include both heavy metals and plant nutrients. This sentence does not make sense logically. Please consider rewriting this sentence with correct information.  
  3. Would you please give out the related references of each number in Table 2?
  4. Line 184-184: “… some beneficial elements (N, P, K, Fe, Zn, Ca+2, Mg+2) in the soil [35]. Table 2 shows that micronutrients concentration (N, P, K, Ca+2) …” Please change Ca+2 and Mg+2 into Ca and Mg, respectively.
  5. Line 248: “…10-20 years if the heavy meatal balances (input from wastewater and output meatal…” spelling mistake “meatal”.
  6. Line 250-265: it is better to move this paragraph about microbiological contamination after the next paragraph about heavy metals’ accumulation (Line 274-287). In addition, the potential treatment methods for microbiological contaminants should be also be proposed.
  7. Interesting literature that show measurements of heavy metals in number of environmental instances could support the manuscript: Lisak Environmental Pollution, 117882 (2021), Bobacka et al Chemosphere 256, 127022 (2020),  Journal of Hazardous Materials 419, 126450 (2021). Citing these will make this manuscript much stronger and vital for whole picture and acceptance.
  8. References: please uniform the format for the references.

Author Response

Review report - water-1462237-peer-review-v1

In the manuscript, "Heavy metals and microbes accumulation in soil and food crops irrigated with wastewater and the potential human health risk: A metadata analysis", the authors conducted a brief but important review covering the effect of wastewater reuse in irrigation on soil, crop (vegetable and forage crops), animal products, and human health. Since wastewater has been widely used for irrigation of vegetables and crops in developing countries, a well-timed review could provide potential readers with the information about assessment and management of risks associated with wastewater irrigation. However, the current review lacks comprehensive, and only a limited number of research works have been cited. Meanwhile, it is also suggested that some contexts should be reorganized more logically. In addition, significant editing of the English is needed. If the revision is performed satisfactorily, the manuscript should be acceptable for publication.

Response: We thank the reviewer comments and suggestion which potentially strengthened the manuscript. In the revised version, 10 new studies have been added included the recommended ones. In addition, the manuscript has been proofread carefully and checked for English grammar and flow.    

Specific comments:

1.Section 2 Wastewater quality for reuse in irrigation of agricultural crops: As wastewater treatment is essential, it is also necessary to review the wastewater treatment methods.

Response: We agree. The wastewater treatment methods has been discussed in the revised version of the manuscript.

2. Line 132-134: “The chemical composition (Ni, Cd, Pb, Fe, N, P, K) of tertiary treated wastewater are often within the World Health Organization (WHO) allowable limits, but heavy metal concentrations have exceeded the maximum limits in several regions of the world, especially, regions that use untreated wastewater for irrigation (Table 2).” Ni, Cd, Pb, Fe, N, P and K include both heavy metals and plant nutrients. This sentence does not make sense logically. Please consider rewriting this sentence with correct information.  

Response: We agree. Text revised.

3.Would you please give out the related references of each number in Table 2?

Response: Table 2 and the related references have been given for each number.

4. Line 184-184: “… some beneficial elements (N, P, K, Fe, Zn, Ca+2, Mg+2) in the soil [35]. Table 2 shows that micronutrients concentration (N, P, K, Ca+2) …” Please change Ca+2 and Mg+2 into Ca and Mg, respectively.

Response: ‘Ca+2’ and ‘Mg+2’ changed into ‘Ca’ and ‘Mg’.

5. Line 248: “…10-20 years if the heavy meatal balances (input from wastewater and output meatal…” spelling mistake “meatal”.

Response: ‘meatal’ changed into ‘metal’.

6. Line 250-265: it is better to move this paragraph about microbiological contamination after the next paragraph about heavy metals’ accumulation (Line 274-287). In addition, the potential treatment methods for microbiological contaminants should be also be proposed.

Response: Paragraph moved and the potential treatment methods for microbiological contaminants have been also be discussed and added to section 2.

7. Interesting literature that show measurements of heavy metals in number of environmental instances could support the manuscript: Lisak Environmental Pollution, 117882 (2021), Bobacka et al Chemosphere 256, 127022 (2020), Journal of Hazardous Materials 419, 126450 (2021). Citing these will make this manuscript much stronger and vital for whole picture and acceptance.

Response: The suggested references has been added to the revised version of the manuscript.

8. References: please uniform the format for the references.

       Response: References revised.

Reviewer 2 Report

General Comment:

I think the main reason for publication of this manuscript in WATER is that the authors have placed this work into a broad setting. In this regard, the literature review does help. However, I think the authors should make it stronger by providing more references in Introduction, especially those are related to human health risk of heavy metals. However, I have added some technical comments that help the authors to improve the scientific merit of this manuscript.

Technical Comments:

ABSTRACT: Please specify the wastewater type? Sanitary? Industry? Agriculture? Or mixture?

L21: Zinc is not a toxic heavy metal. Please revise.

L21-23: Microbial count (total coliforms, fecal coliforms, Escherichia coli) in untreated wastewater are always higher than the world standard limits. This is not a finding. Please remove.

ABSTRACT: Please use capital letters for “cfu”.

ABSTRACT: Do you mean carcinogenic or non-carcinogenic risk? Please specify.

ABSTRACT: Please add the number of studies used for metadata analysis.

INTRODUCTION: Again, carcinogenic or non-carcinogenic risk? Please specify.

L56-57: Please support this important statement with larger scale studies such as “Iran’s groundwater hydrochemistry” and “Iran’s agriculture in the Anthropocene”.

INTRODUCTION: Did you read the references paper in your manuscript’s subject? Did you read Thebo’s paper published in Environ Res Letters?

INTRODUCTION: Please add a Table to supplementary materials and simply give information about studies (more than 70) used in the manuscript (e.g., title, year, publication name, …).

Table 1: According to a study conducted by Thebo et al. (2017), Iran was ranked among the countries with the highest irrigated croplands with untreated effluents, along with China, India, Mexico, and Pakistan. Why did you exclude Iran from your study? Please explain.

Table 1: What do you mean by “Long-term average annual precipitation in depth”?

Table 1: Can you separate rain-fed and irrigated cultivated areas?

Did you produce Figure 1 by yourself? If else, you need to get permission for this figure.

Table 2: unit of TC, FC, and E. coli is MPN. But, you addressed their unit as “CFU” in Abstract. Please explain or simply revise it.  

Table 2: What do you mean by phosphate? Orthophosphate or total phosphate? Please specify.

HUMAN HEALTH RISK ASSESSMENT OF WASTEWATER-IRRIGATED CROPS: Again, carcinogenic or non-carcinogenic risk? Please specify (see and cite “Alarming carcinogenic and non-carcinogenic risk of heavy metals in Sabalan dam reservoir, Northwest of Iran” and “Metal contamination assessment in water column and surface sediments of a warm monomictic man-made lake: Sabalan Dam Reservoir, Iran”.

Author Response

Comments and Suggestions for Authors

General Comment:

I think the main reason for publication of this manuscript in WATER is that the authors have placed this work into a broad setting. In this regard, the literature review does help. However, I think the authors should make it stronger by providing more references in Introduction, especially those are related to human health risk of heavy metals. However, I have added some technical comments that help the authors to improve the scientific merit of this manuscript.

Technical Comments:

ABSTRACT: Please specify the wastewater type? Sanitary? Industry? Agriculture? Or mixture?

Response: In most developed countries there is no protocol to separate the wastewater sources, therefore the source of wastewater is a mixture of residential, industrial and agricultural.

L21: Zinc is not a toxic heavy metal. Please revise.

Response: We agree. Text revised.

L21-23: Microbial count (total coliforms, fecal coliforms, Escherichia coli) in untreated wastewater are always higher than the world standard limits. This is not a finding. Please remove.

Response: We agree. The sentence removed.

ABSTRACT: Please use capital letters for “cfu”.

Response: We agree. ‘cfu’ capitalized.

ABSTRACT: Do you mean carcinogenic or non-carcinogenic risk? Please specify.

Response: The critical problem is the carcinogenic risk. Text revised.

ABSTRACT: Please add the number of studies used for metadata analysis.

Response: We agree. The number of studies used for metadata analyses has been added. 

INTRODUCTION: Again, carcinogenic or non-carcinogenic risk? Please specify.

Response: Text revised.

L56-57: Please support this important statement with larger scale studies such as “Iran’s groundwater hydrochemistry” and “Iran’s agriculture in the Anthropocene”. INTRODUCTION: Did you read the references paper in your manuscript’s subject? Did you read Thebo’s paper published in Environ Res Letters?

Response: We agree. The study by Thebo which published in Env. Res. Lett. Added to the revised version of the manuscript.

INTRODUCTION: Please add a Table to supplementary materials and simply give information about studies (more than 70) used in the manuscript (e.g., title, year, publication name, …).

Response: I’m not sure what the reviewer mean by this comments. The title, year and publication name are all listed in the reference section.

Table 1: According to a study conducted by Thebo et al. (2017), Iran was ranked among the countries with the highest irrigated croplands with untreated effluents, along with China, India, Mexico, and Pakistan. Why did you exclude Iran from your study? Please explain.

Response: Good point. Iran statistics added to the revised version of the manuscript.

Table 1: What do you mean by “Long-term average annual precipitation in depth”?

Response: Text revised.

Table 1: Can you separate rain-fed and irrigated cultivated areas?

Response: Cultivated area separated to rain-fed and irrigated area and latest statistics by FAO has been used in the revised version of the manuscript. 

Did you produce Figure 1 by yourself? If else, you need to get permission for this figure.

Response: Yes, we produce this figure by our self.

Table 2: unit of TC, FC, and E. coli is MPN. But, you addressed their unit as “CFU” in Abstract. Please explain or simply revise it.  

Response: We agree. Table 2 units revised.

Table 2: What do you mean by phosphate? Orthophosphate or total phosphate? Please specify.

Response: We mean Orthophosphate, text revised.

HUMAN HEALTH RISK ASSESSMENT OF WASTEWATER-IRRIGATED CROPS: Again, carcinogenic or non-carcinogenic risk? Please specify (see and cite “Alarming carcinogenic and non-carcinogenic risk of heavy metals in Sabalan dam reservoir, Northwest of Iran” and “Metal contamination assessment in water column and surface sediments of a warm monomictic man-made lake: Sabalan Dam Reservoir, Iran”.

 Response: We agree. Both studies results has been added to the revised version of the manuscript.

Round 2

Reviewer 1 Report

Authors addressed comments.

Reviewer 2 Report

Thanks to the Authors. All my comments have been responded properly.

Good luck